# Improvement in quality of life and cognitive function in Post-COVID syndrome after online occupational therapy: Results from a randomized controlled pilot study

Dominik Schröder[1], Andrea Stölting[2]*, Christina Müllenmeister[1],
Georg M. N. Behrens[2,3], Sandra Klawitter[4,5], Frank Klawonn[4,5], Aisha Cook[6],
Nadja Wegner[2], Martin Wetzke[7], Tim Schmachtenberg[1],
Alexandra Dopfer-Jablonka[2,3‡], Frank Müller[1,8‡], Christine Happle[3,7,9‡]

1 Department of General Practice, University Medical Center Göttingen, Göttingen, Germany,
2 Department of Rheumatology and Immunology, Hannover Medical School, Hannover, Germany,
3 German Center for Infection Research, RESIST, Hannover-Braunschweig, Germany, 4 Institute of Information Engineering, Ostfalia University of applied sciences, Wolfenbüttel, Germany, 5 Biostatistical Research Group, Helmholtz-Center for Infection Research, Braunschweig, Germany, 6 Tim-Cook Occupational Therapy, Hannover, Germany, 7 Department of Pediatric Allergology, Pneumology, and Neonatology, Hannover Medical School, Hannover, Germany, 8 Department of Family Medicine, College of Human Medicine, Michigan State University, Grand Rapids Michigan, United States of America,
9 German Center for Lung Research, DZL-BREATH, Hannover, Germany

☯ These authors contributed equally to this work.
‡ AD-J, FM and CH also contributed equally to this work.
* stoelting.andrea@mh-hannover.de

## Abstract

### Background

Post-COVID syndrome (PCS) poses enormous clinical challenges. Occupational therapy (OT) is recommended in PCS, but structural validation of this concept is pending.

### Methods

In an unblinded randomized pilot study (clinical trial # DRKS0026007), feasibility and effects of online OT in PCS were tested. Probands received structured online OT over 12 weeks either via interactive online treatment sessions (interactive group) or prerecorded videos (video group). 50% of probands received no online OT (control group). At week 0, 12, and 24, we analyzed study experience, health-related quality of life, cognitive functions. impairment in performance, and social participation.

### Results

N = 158 probands (mean age 38 yrs., 86% female) were included into the analyses. The study experience was described as positive or very positive in 83.3% of

**Data availability statement:** Data sets for this study are not publicly available due to a decision of the responsible research ethics board. In particular, the ethics vote requires us to be able to remove personal data at the request of our participants in the future. Data may however be shared upon reasonable request within a formal data sharing agreement. Request for the data may be sent to the research groups assistant who is responsible for communication with interested parties (Mail to: Ignacio.Melanie@mh-hannover.de; Melanie Ignacio, Dept. for Immunology and Rheumatology, Carl-Neuberg-Strasse 1, 30625 Hannover). M. Ignacio has no direct relationship to the data set or research presented in the paper.

**Funding:** The project received funding through the German Federal Ministry for Education and Research (Grant number 01EP2103C). The funders had no role in study design, data collection and analysis, decision to publish, or preparation of the manuscript.

**Competing interests:** The authors have read the journal's policy and have the following competing interests: Dominik Schröder is a paid employee of Pfizer Deutschland GmbH. This does not alter our adherence to PLOS ONE policies on sharing data and materials.

probands in the interactive versus 48.1% of probands in the video group (p = 0.001). After 12 weeks, all groups displayed significant improvement in concentration, memory, and performance of daily tasks. After 24 weeks, significant improvement in concentration and memory were observed in control- and video-probands, and social participation had improved after video-OT. However, only probands in the interactive online OT group showed improvement of all measured endpoints including concentration, memory, quality of life, and social participation.

## Conclusion

We show that online OT is feasible, and that interactive online OT is a promising treatment strategy for affected patients. We present exploratory data on its efficacy and describe variables that can be employed for further investigations in confirmatory trials.

---

## Introduction

Approximately 3–10% of people with a severe acute respiratory syndrome coronavirus type 2 (SARS-COV-2) infection develop Post-COVID syndrome (PCS) with ongoing symptoms [1–3]. Definitions of PCS vary but are characterized by symptoms and/or delayed or long-term complications, persisting or beginning beyond four weeks after a SARS-CoV-2 infection, and PCS affected people often experience severe fatigue, trouble concentrating, and reduced quality of life and social participation [4–6]. Current therapy guidelines for PCS mention occupational therapy (OT) as a treatment option [6], and case reports describe success of OT in treating this novel condition [7]. However, clinical studies evaluating the efficacy of OT in PCS are pending.

We sought to combine the concept of OT for PCS with the strategy of remote (digital) treatment delivery. Remote treatment strategies via digital communication have become more accepted since the surge of Corona Virus Disease 2019 (COVID-19) [8]. As many PCS patients suffer from reduced social participation, impaired mobility, chronic fatigue, and problems to structure their day, digital treatment with low threshold to participate appears to be particularly appealing for this patient group [9]. Given the high prevalence of PCS, the scalability of digital therapy formats could enable efficient treatment for a high number of affected people.

Here, we tested the feasibility and explored the efficacy of structured online OT for PCS with regard to its impact on cognitive function, social participation, and quality of life in a randomized controlled pilot study (German clinical trial registry #DRKS00026007 [10]). The improvement of outcomes was measured by standardized and scientifically validated analysis tools. The OT intervention was based on a detailed manual tailored to the needs of PCS patients and delivered online, either by interactive sessions or as prerecorded videos. Alongside with description of acceptance and feasibility of online OT, the objective of the here presented pilot study was to evaluate the hypothesis that this treatment approach could improve major PCS symptoms such as fatigue and cognitive impairments. The OT intervention should tailor the needs of affected people with PCS and improve their quality of life and social participation.

## Methods

### Trial design

This is a randomized controlled pilot study with two interventional groups receiving OT either via (I) interactive digital sessions (interactive group) or (II) prerecorded videos (video group). A control group (III) received no intervention (controls). Planned allocation ratio was 1:1:2 (I:II:III). Patients were evaluated upon study start (T1), after the 12-week treatment phase (T2) and 24 weeks after study start (T3). OT was delivered in units of 24 standardized sessions (30 minutes long each) which were delivered twice a week. OT included specific instructions for managing PCS symptoms, e.g., through breathing exercises in case of fear and stress symptoms or pacing maneuvers in case of recurring brain fog and fatigue, and education on symptom pathology and relief strategies. By assessing the participants individual needs (in interviews or written exercises) and re-assessing these needs and symptoms throughout the intervention period, patients were able to focus on their most immanent complaints. Participants received a dedicated workbook with exercises and materials to support the application of relieving techniques into their everyday life. More study details are outlined in a study protocol [10].

### Participants and recruitment

Following inclusion criteria were applied [1]: age ≥ 16 years [2], persistent or new PCS symptoms ≥ 4 weeks after SARS-CoV-2 infection (confirmed by PCR or rapid antigen testing) [3], feeling of strong cognitive impairment and/or fatigue (concentration deficits and/or fatigue) ≥ 5/10 on a Likert scale [4], access to a digital device [5], consent to participate in the study. We performed recruitment through an online study platform for persons affected by PCS (DEFEAT Corona, German study registry number DRKS00026007 [11]). Interested patients were asked to complete an online screening survey. No specific educational level was necessary to enter the study, but the inclusion survey and all other study materials and communication were provided in German language only, hence fluency in German language was a prerequisite to enter the study. Exclusion criteria were refusal or inability to declare consent and the lack of willingness or ability to complete the program. PCS people meeting the inclusion criteria were invited to participate by email. Prior to enrollment, individual consent interviews were conducted which each interested eligible person and further study information was provided. Participants declared their written consent online before enrollment.

### Interventions

Participants in the interactive group received online OT by an experienced occupational therapist twice weekly through interactive digital sessions. Probands in the video group were provided with links for prerecorded OT videos for two weekly sessions. Outline and structure of OT were based on a detailed manual tailored to the needs of PCS patients, which was identical in both interventional groups [10]. OT sessions consisted of guided exercises to control typical PCS symptoms, as well as customizable units to improve performance in everyday occupations, social participation, and wellbeing. For example, to reflect on personal hurdles and sustainability factors, participants were asked to visualize their major challenges in everyday occupations in a drawing or to list factors that they experienced to be beneficial for their relaxation or wellbeing. Participants in both groups also received OT workbooks and were asked to apply contents into their everyday life complementing the OT content of each therapy session regularly. Control participants received no treatment.

### Outcomes

Feasibility and acceptance of the treatment concept (primary endpoints) were analyzed by Likert scaled items (online questionnaire, patient reported measures (PROMs)) and assessment of dropout rates in the different study phases. Secondary endpoints were cognitive function and problems in everyday occupations assessed by Neuro-QoL™ (NeuroQuality of Life - self-report on health-related quality of life in people with neurological disorders v2.0 cognitive function short form)

[12], memory function tested by the WIT-2 (Wilde Intelligenztest-2) tool [13], and concentration deficits evaluated by the d2-R test [14]. Health-related quality of life was analyzed by the EQ-5D-3L index and EQ VAS (allow self-assessment of health status based on 5 dimensions: mobility, self-care, daily activities, pain, and anxiety) [15]. Social participation was evaluated by the Index for measuring participation restriction (IMET) score [16], and occupational performance in interviews employing the Canadian occupational performance measure (COPM) [13]. All questionnaires and assessments were provided and/or conducted in German language.

## Sample size

Considering feasibility and project funding of this pilot study, we conceived sample sizes pragmatically. We planned to recruit 160 participants, 80 of whom were randomized to the control group, 40 to the video intervention and 40 to the interactive group.

## Randomization, testing for occupational problems

After cognitive function testing, probands were randomized into the three groups employing an urn model (ratio interactive/video/control group: 1/1/2) by a member of the study team not involved in the treatment intervention. Patients were then tested for occupational problems using a structured interview according to the Canadian occupational performance measure (COPM [13]), again by an independent team member not involved in the intervention.

## Statistical analysis

All analyses were performed using R (version 4.2.3). Proband characteristics were compared via Kruskal–Wallis tests (continuous variables) and Fisher–Freeman–Halton tests (categorical variables). Because within-group outcomes did not meet normality assumptions, we applied Wilcoxon signed-rank tests (Holm-adjusted) for changes over time (T1, T2 & T3) within each study arm. Median and Q25-Q75 were reported for each study arm and time point. For the between-group analyses, we followed an intention-to-treat (ITT) approach, using multiple imputations by chained equations (m = 5) to address missing data (mice package [17]). Multiple imputations on outcomes were performed using additional baseline proband characteristics (sex, age, fatigue score and trouble concentrating score). Imputed datasets were analyzed and pooled with Rubin's rules [18]. In each imputed dataset, we fit a linear mixed-effects model (LMM) with fixed effects for time, intervention arm, their interaction, a restricted cubic spline for baseline values (df = 3 [19]), and a random intercept per participant. Pairwise contrasts were extracted (emmeans package [20]), pooled, and adjusted for multiple testing (Holm's method). Additionally, a per-protocol analysis was performed using repeated-measures ANOVA (afex package [21]) on the subset with missing data excluded. Prior to performing the repeated-measures ANOVA, the underlying assumptions were assessed for compliance. Between-group analyses were reported with mean and 95% CI. All tests were two-sided with $p < 0.05$ considered significant.

## Research ethics

The study protocol was approved by responsible research ethics boards of all participating centers (Hannover Medical School #9948_BO_K_2021, University Medical Center Göttingen 15/8/22Ü). The study was registered at the German Registry for Clinical Trials (trial number DRKS00026007). Participants provided written informed consent (online through digital signature) prior to enrollment. The project received funding through the German Federal Ministry for Education and Research (Grant number 01EP2103C).

Data sets for this study are not publicly available due to a decision of the responsible research ethics board. In particular, the ethics vote requires us to be able to remove personal data at the request of our participants in the future. Data may however be shared upon reasonable request within a formal data sharing agreement. Request for the data may be

sent to the research groups assistant who is responsible for communication with interested parties (Mail to: Ignacio.Melanie@mh-hannover.de; Melanie Ignacio, Dept. for Immunology and Rheumatology, Carl-Neuberg-Strasse 1, 30625 Hannover). M. Ignacio has no direct relationship to the data set or research presented in the paper.

## Results

Initially, n = 163 PCS patients were recruited, but n = 5 had to be excluded secondarily due to failing the inclusion criteria (Fig 1). In total, n = 158 PCS patients were included in the final analysis. Participant numbers for each study arm and time-point are illustrated in Fig 1.

   From cases in the interventional arms with follow-up data, information on subjective treatment effects was available for 98.7% of probands, and 72.2% provided information on their personal OT experience. Psychological testing was performed in 98.1%, and COPM in 88% of probands with follow up evaluation.

### Participant characteristics

The median age of participants was 38 years (IQR 30–45, range 16–67), and probands in the interventional groups were significantly younger than those in the control group (p = 0.04). The majority of participants (86.1%) were female, with

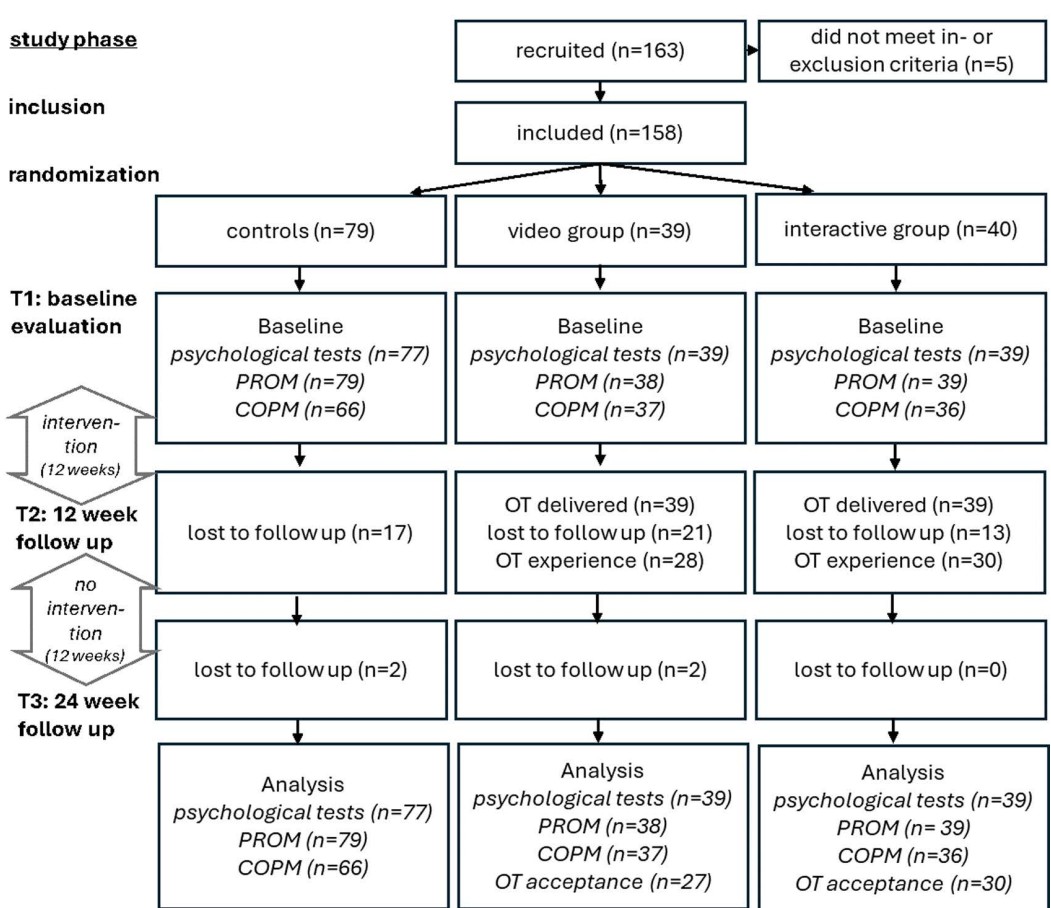

**Fig 1. Flowchart illustrating study phases and proband numbers (PROM: patient reported outcome measures include NeuroQoL, EQ-5D, IMET).**

more women in the interventional as in control groups (p = 0.02). PCS participants in the interactive group presented with better memory performance (WIT-2) than those in control and video arms (p = 0.01). All further baseline variables were not significantly different across study arms and are displayed in Table 1. Further information on sociodemographic characteristics is provided in S1 Table.

## Participation and intervention acceptance

To analyze acceptance and feasibility of our concept, we analyzed OT treatment continuation and asked PCS patients to evaluate their OT experience during the 12-week intervention phase. 65.2% of participants completed the 12-week interventional or control period and were available for evaluation upon timepoint T2. 32.5% of probands in the interactive, and 59.0% of participants in the video group were lost to follow up at T2, as were 24.1% of control probands. An additional 2.5% of all remaining participants were lost to follow up at timepoint T3. In an online survey, all participants of the interventional arms could rate their therapy perception between 0 (very negative) and 5 (very positive). Participants in the interactive group evaluated their OT experience with a mean of 4.2 points (range 2–5, n = 30), and 83.3% rated the treatment as positive or very positive (Fig 2). By contrast, probands in the video group rated their OT experience with an average of

**Table 1. Proband characteristics at baseline.**

| | n | Total (N = 158) | Control (N = 79) | Video (N = 39) | Interactive (N = 40) |
|---|---|---|---|---|---|
| Age (years)[a] | 158 | 38.0 (30.0–45.0) Range 16–67 | 42.0 (31.5–47.0) Range 16–61 | 35.0 (26.5–41.0) Range 18–53 | 36.0 (29.0–44.0) Range 16–67 |
| Sex | 158 | | | | |
| Male[b] | | 22 (13.9) | 16 (20.3) | 1 (2.6) | 5 (12.5) |
| Female[b] | | 136 (86.1) | 63 (79.7) | 38 (97.4) | 35 (87.5) |
| SARS-CoV-2 infection until study start (weeks)[a] | 153 | 41.1 (26.8–63.9) | 43.1 (28.1–60.9) | 38.0.7 (20.9–51.2) | 42.6 (28.9–98.3) |
| Fatigue[a] (Likert x/10) | 158 | 8.0 (7.0–9.0) | 8.0 (7.0–9.0) | 9.0 (7.0–10.0) | 8.0 (7.0–9.0) |
| Trouble concentrating (Likert x/10)[a] | 158 | 8.0 (7.0–9.0) | 8.0 (7.0–9.0) | 8.0 (7.0–9.0) | 7.0 (6.0–9.0) |
| d2-R[a] | 155 | -1.0 (-1.6 – -0.2) | -1.1 (-1.0 – -0.2) | -0.9 (-1.6 – -0.2) | -0.8 (-1.5 – -0.1) |
| WIT-2[a] | 155 | 0.1 (-0.6–0.5) | -0.1 (-0.5–0.3) | -0.1 (-1.0–0.3) | 0.5 (-0.2–0.9) |
| EQ-5D-3L Index[a] | 156 | 0.61 (0.39–0.75) | 0.61 (0.39–0.75) | 0.61 (0.38–0.75) | 0.61 (0.484–0.75) |
| EQ VAS[a] | 156 | 36.0 (30.0–51.0) | 37.0 (30.0–50.0) | 35.5 (30.3–49.0) | 36.0 (31.0–56.0) |
| Neuro-QoL™ v2.0 Cognitive Function-Short Form[a] | 155 | 33.0 (29.8–36.0) | 34.0 (29.8–36.0) | 33.0 (28.6–37.0) | 32.0 (28.6–35.0) |
| IMET[a] | 152 | 55.0 (41.3–66.0) | 53.5 (41.0–66.0) | 54.5 (44.8–66.0) | 59.5 (37.8–66.0) |
| COPM – satisfaction[a] | 139 | 2.3 (1.5–3.0) | 2.3 (1.5–3.0) | 2.5 (1.3–3.0) | 2.4 (1.5–3.3) |
| COPM – performance[a] | 139 | 3.0 (2.5–3.6) | 3.0 (2.5–3.5) | 3.0 (2.5–3.8) | 3.0 (2.5–3.8) |

Demographic and baseline phenotyping in the three experimental groups (Control = participants without intervention, Video = participants with pre-recorded video interventions, Interactive = participants with bi-weekly personal occupational therapy via messenger). Cognitive impairments were assessed using the d2-R (Deficits in concentration, attention, and mental speed) and the WIT-2 (Wilde Intelligenztest). EQ-5D-3L and EQ VAS are questionnaires from the EuroQoL Group (Quality of Life), which allow self-assessment of health status based on 5 dimensions: mobility, self-care, daily activities, pain, and anxiety. The Neuro-QoL is a self-report on health-related quality of life in people with neurological disorders. 3L describes the state of health divided into three response levels. The IMET (Index zur Messung von Einschränkungen der Teilhabe) can measure participation and involvement in patients with various chronic conditions. The COPM (Canadian Occupational Performance Measure) provides metrics for changes in occupational performance, satisfaction, and significance, and improves the transparency of OT outcomes.

[a]median (Q25-Q75);

[b]number (proportion).

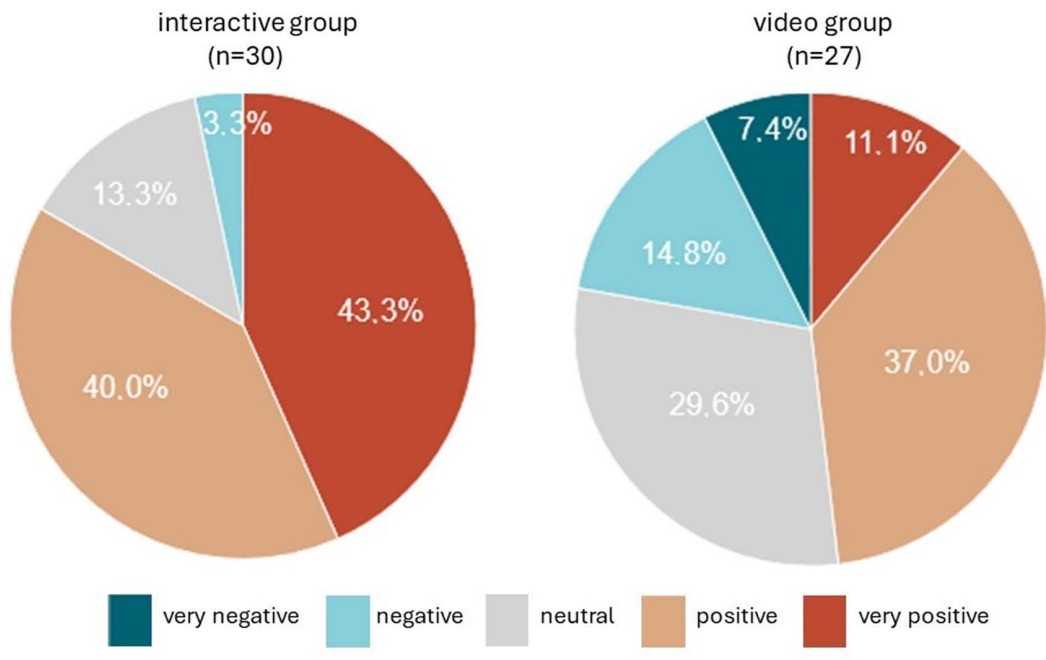

**Fig 2. Rating of study experience by participants in the two intervention groups.**

3.3/5 points (range 1–5, n = 27), and only 48.1% described their experience as positive or very positive (Fig 2, difference in satisfaction p = 0.001).

### Differences between groups

To test our hypothesis that online OT could improve PCS symptoms, we performed an exploratory analysis of the intervention's efficacy in improving factors such as quality of life and fatigue. Compared with the control group, the interactive group displayed significant improvements in health-related quality of life (EQ-5D-3L index, mean difference 0.11, 95% CI [0.05–0.17], p = 0.003) and cognitive function (Neuro-QoL, mean difference 3.95, 95% CI [2.04–5.86], p < 0.001, Table 2) between baseline and first follow-up. The EQ-VAS measuring self-assessed health status was significantly improved in PCS patients after interactive OT (mean difference 8.04, 95% CI [2.27–13.80], p = 0.038). In this group, we also observed a significant amelioration of health-related quality of life upon final assessment (EQ-5D-3L index mean difference 0.09, 95% CI [0.03–0.15], p = 0.026). For the other endpoints, there were no significant differences between the study groups (Table 2). Excluding cases with missing data (rather than using MI as presented in Table 2) did not change significance regarding the outcomes health-related quality and cognitive function upon first follow-up (S1 Table).

### Longitudinal differences within groups

In all groups, significant improvements in concentration and memory function, as well as occupational performance were observed during the first 12 weeks of the study. However, only in participants after interactive online OT, a significant improvement in health-related quality of life, social participation, and cognitive function as assessed by Neuro-QoL occurred (Table 3).

24 weeks after the start of the study, all groups displayed significant improvement in concentration and memory performance, and participants in the control and interactive groups showed significantly enhanced social participation, but only

**Table 2. Differences between study arms adjusted for baseline scores (multiple imputations).**

| | T2 difference (95% CI)[1] | T3 difference (95% CI)[1] | P T2[2] | P T3[2] |
|---|---|---|---|---|
| **d2-R[3]** | | | | |
| video vs. control | -0.01 (-0.27–0.26) | -0.10 (-0.36–0.16) | 1.00 | 1.00 |
| interactive vs. control | 0.16 (-0.11–0.42) | -0.07 (-0.33–0.19) | 1.00 | 1.00 |
| **WIT-2[3]** | | | | |
| video vs. control | 0.18 (-0.11–0.46) | -0.36 (-0.64 – -0.08) | 0.663 | 0.066 |
| interactive vs. control | 0.29 (0.00–0.57) | -0.10 (-0.38–0.19) | 0.233 | 0.986 |
| **EQ-5D-3L index[3]** | | | | |
| video vs. control | 0.01 (-0.05–0.07) | -0.01 (-0.07–0.05) | 1.00 | 1.00 |
| interactive vs. control | 0.11 (0.05–0.17) | 0.09 (0.03–0.15) | 0.003 | 0.026 |
| **EQ VAS[3]** | | | | |
| video vs. control | 2.27 (-3.55–8.10) | 4.23 (-1.59–10.10) | 0.808 | 0.463 |
| interactive vs. control | 8.04 (2.27–13.80) | 7.10 (1.34–12.90) | 0.038 | 0.079 |
| **Neuro-QoL™ v2. 0 cognitive function short form[3]** | | | | |
| video vs. control | -0.17 (-2.06–1.72) | -1.26 (-3.16–0.63) | 0.859 | 0.435 |
| interactive vs. control | 3.95 (2.04–5.86) | 1.42 (-0.49–3.34) | <0.001 | 0.435 |
| **IMET[4]** | | | | |
| video vs. control | 0.78 (-4.26–5.83) | -3.78 (-8.83–1.26) | 1.00 | 0.565 |
| interactive vs. control | -6.62 (-11.70 – -1.54) | -3.01 (-8.09–2.06) | 0.064 | 0.734 |
| **COPM performance[4]** | | | | |
| video vs. control | 0.13 (-0.13–0.38) | – | 1.00 | – |
| interactive vs. control | 0.15 (-0.11–0.40) | – | 1.00 | – |
| **COPM satisfaction[4]** | | | | |
| video vs. control | 0.32 (0.01–0.63) | – | 0.243 | – |
| interactive vs. control | 0.04 (-0.28–0.35) | – | 1.00 | – |

The table shows mean differences between the three experimental groups (Control = participants without intervention, Video = participants with pre-recorded video interventions, Interactive = participants with bi-weekly personal occupational therapy via messenger). Cognitive impairments were assessed using the d2-R (Deficits in concentration, attention, and mental speed) and the WIT-2 (Wilde Intelligenztest). EQ-5D-3L and EQ VAS are questionnaires from the EuroQoL Group (Quality of Life), which allow self-assessment of health status based on 5 dimensions: mobility, self-care, daily activities, pain, and anxiety. The Neuro-QoL is a self-report on health-related quality of life in people with neurological disorders. The IMET (Index zur Messung von Einschränkungen der Teilhabe) can measure participation and involvement in patients with various chronic conditions. The COPM (Canadian Occupational Performance Measure) provides metrics for changes in occupational performance, satisfaction, and significance, and improves the transparency of OT outcomes. [1]differences and CI adjusted according to baseline values using a linear mixed-effect model, [2]Bonferroni Holm adjusted for multiple testing, [3]positive value: improvement, [4]negative value: improvement, -: not assessed.

PCS patients in the interactive group presented with significant improvement of all measured end points such as cognitive function, social participation, health-related quality of life, and occupational performance over time (Table 3)

## Discussion

PCS is a growing global health problem which has been estimated to affect up to 11% of patients after SARS-CoV-2 infection [3]. In spite of high patient numbers worldwide and strong clinical need, effective treatment strategies are pending [22,23]. PCS associated symptoms such as fatigue and cognitive impairment negatively impact occupational performance and significantly affect satisfaction across multiple domains, such as the ability to work and navigate everyday life [24,25].

Here, we present first results on the feasibility of online OT and explore it´s efficacy in PCS. Our data illustrates that online OT can be a valuable treatment approach in helping PCS patients. We show that online delivered OT was accepted

**Table 3. Longitudinal changes within groups of cognitive performance, quality of life, and occupational performance.**

| | T1[1] (Baseline) | T2[1] (12-Weeks) | T3[1] (24-Weeks) | T1-T2[2] p | P T2-T3[2] p | P T1-T3[2] p |
|---|---|---|---|---|---|---|
| **Control** | | | | | | |
| d2-R (n=77) | -1.10 (-1.90 – -0.20) | -0.50 (-1.50–0.30) | -0.30 (-1.20–0.30) | <0.001 | 0.002 | <0.001 |
| WIT-2 (n=77) | -0.10 (-0.50–0.30) | 0.30 (-0.30–1.00) | 1.00 (-0.10–1.60) | <0.001 | <0.001 | <0.001 |
| EQ-5D-3L Index (n=79) | 0.613 (0.386–0.750) | 0.613 (0.381–0.750) | 0.649 (0.391–0.757) | 0.912 | 0.092 | 0.419 |
| EQ VAS (n=79) | 37.00 (30.00–50.00) | 36.00 (30.00–55.00) | 35.00 (26.50–63.00) | 0.590 | 0.590 | 0.590 |
| Neuro-QoL™ v2. 0 cogn. function-short form (n=79) | 34.00 (29.80–36.00) | 32.00 (28.60–37.00) | 35.00 (29.80–38.90) | 0.145 | <0.001 | 0.074 |
| IMET (n=78) | 53.50 (41.00–66.00) | 53.00 (37.00–66.75) | 52.00 (31.00–66.50) | 0.125 | 0.125 | 0.021s |
| COPM performance (n=66) | 3.00 (2.50–3.50) | 3.50 (2.54–4.50) | – | <0.001 | – | – |
| COPM satisfaction (n=66) | 2.25 (1.50–3.00) | 2.50 (1.54–3.50) | – | <0.001 | – | – |
| **Video** | | | | | | |
| d2-R (n=39) | -0.90 (-1.60– -0.15) | -0.60 (-1.35–0.30) | -0.40 (-1.35–0.30) | 0.003 | 0.061 | 0.002 |
| WIT-2 (n=39) | -0.10 (-1.00–0.30) | 0.10 (-0.30–1.15) | 0.30 (-0.60–1.60) | 0.002 | 0.495 | 0.001 |
| EQ-5D-3L Index (n=38) | 0.613 ( 0.38–0.75) | 0.639 (0.381–0.750) | 0.64 (0.39–0.77) | 1.00 | 1.00 | 1.00 |
| EQ VAS (n=38) | 35.50 (30.25–49.00) | 35.50 (26.75–50.00) | 36.50 (26.75–51.25) | 0.990 | 0.460 | 0.990 |
| Neuro-QoL™ v2. 0 cogn. function-short form (n=37) | 33.00 (28.60–37.00) | 33.00 (39.80–37.00) | 35.00 (27.30–38.90) | 1.00 | 0.200 | 1.00 |
| IMET (n=36) | 54.50 (44.75–66.00) | 56.50 (44.75–64.25) | 55.50 (39.75–64.25) | 0.840 | 0.640 | 0.640 |
| COPM performance (n=37) | 3.00 (2.50–3.75) | 3.50 (3.00–4.75) | – | 0.001 | – | – |
| COPM satisfaction (n=37) | 2.50 (1.25–3.00) | 2.75 (2.25–4.75) | – | <0.001 | | |
| **Interactive** | | | | | | |
| d2-R (n=39) | -0.80 (-1.50 – -0.10) | -0.10 (-0.95–0.80) | -0.20 (-0.95–0.80) | <0.001 | 0.002 | <0.001 |
| WIT-2 (n=39) | 0.50 (-0.20–0.85) | 1.00 (0.10–1.60) | 1.30 (0.30–1.90) | <0.001 | 0.028 | <0.001 |
| EQ-5D-3L Index (n=38) | 0.613 (0.484–0.754) | 0.750 (0.5–0.79 | 0.750 (0.57–0.79) | 0.044 | 0.346 | 0.008 |
| EQ VAS (n=38) | 36.00 (31.00–56.00) | 46.00 (34.00–65.00) | 42.00 (32.50–65.00) | 0.047 | 0.979 | 0.047 |
| Neuro-QoL™ v2. 0 cogn. function-short form (n=37) | 32.00 (28.60–35.00) | 34.00 (30.90–37.45) | 34.00 (30.35–39.40) | 0.026 | 0.132 | 0.004 |
| IMET (n=36) | 59.50 (37.75–66.00) | 48.50 (36.75–64.00) | 49.00 (36.00–66.00) | 0.023 | 0.836 | 0.012 |
| COPM performance (n=36) | 3.00 (2.50–3.75) | 3.88 (2.94–4.81) | – | 0.001 | – | – |

*(Continued)*

**Table 3.** (Continued)

| | T1[1] (Baseline) | T2[1] (12-Weeks) | T3[1] (24-Weeks) | T1-T2[2] p | P T2-T3[2] p | P T1-T3[2] p |
|---|---|---|---|---|---|---|
| **COPM satisfaction (n = 36)** | 2.38 (1.46–3.25) | 3.00 (1.63–4.50) | – | 0.029 | – | – |

[1]Median (Q25-Q75);

[2]Wilcoxon sign rank, Bonferroni Holm correction for multiple testing, -: not assessed at T3.

by the majority of PCS patients participating in our study. Although we only present exploratory data on the therapeutic effect of our online OT intervention, our data support the notion that online OT can significantly improve cognitive function and quality of life in people with PCS. Importantly, the presented outcome parameters can be employed in larger randomized controlled clinical trials on novel PCS treatment strategies.

Online delivered OT has several advantages especially for people with PCS: It can easily be scaled to meet the currently high demand in PCS treatment. It may also be especially suitable for PCS people that struggle with impaired mobility, reduced social participation, chronic fatigue, or live in more remote places. In our study, online provision of OT was used by participants of all ages, illustrating the wide range of PCS patients that could be reached by such an approach.

Our results suggest that online OT delivered through interactive "person to person" contact may be accepted particularly well in PCS treatment. Around two thirds of PCS patients in the interactive therapy group completed the 12-week treatment course in the interactive group as compared to less than one half of those in the video group, and significantly more patients that were treated interactively rated their OT experience as positive. Furthermore, PCS patients after interactive OT displayed significantly stronger treatment benefits when compared to persons from the video and control groups. Only patients in the interactive treatment group showed significant improvement of all measured endpoints of cognitive function, occupational problems, quality of life, and social participation. Their cognitive function showed statistically significant improvement and their health-related quality of life had increased by 0.11 points, a value clearly above the threshold of 0.03 previously described to be clinically relevant [26]. The fact that secondary outcomes such as memory and concentration capacity also improved in the control group could be explained by a natural amelioration of PCS symptoms over time, which has also been observed by other researchers [27].

Thus far, little is known on the pathophysiology of PCS, but cumulating evidence suggests fundamental changes in brain function and neurotransmission [28,29]. Different novel non-pharmaceutical treatment options targeting neurocognitive problems in PCS have been suggested, such as psychotherapy or active transcranial current stimulation [30–32]. The latter approach was reported to be associated with a statistically significant improvement in fatigue when compared with sham stimulation, however, case numbers (n = 23 treated patients) were low [32]. Future studies of us and others will focus on assessing how neurocognitive problems in PCS can be addressed effectively.

Our study has several limitations. Firstly, due to its novelty, *a priori* power calculations could not be performed and the number of enrolled participants only allowed for exploratory analysis of treatment effects. Also, the inhomogeneity of groups upon study starts with regard to age, sex, and memory function may have impacted our results, and we plan to conduct a larger study to analyze the efficacy of online OT in patients in sex- and age- specific or other subgroups. The high rate of lost to follow up participants needs to be addressed in future studies. Unfortunately, we were unable to systematically assess reasons for study discontinuation in our cohort, which will be a focus in our follow up studies. The high proportion of female study participants may have biased our results. But this phenomenon was also observed in other clinical PCS-studies [32] and may be explained by a higher rate of female PCS patients in general and the fact that women tend to seek medical help earlier than men [26,27]. Furthermore, central outcomes such as symptom severity were self-reported. As our pilot study was performed in unblinded fashion, this could have biased self-assessment. Some

of our test instruments such as WIT-2 and d2r are not validated for repeated measurements, which could have led to improvement of scorings at the second test date. As the use of the statistical method LOCF is controversial, we included results without this approach to the manuscripts supplement, showing that no significant differences with and without this approach occur. Lastly, in contrast to other clinical endpoints, patients were for occupational problems by COPM [13] shortly *after* randomization, but before starting the intervention. This was due to technical reasons, but may have affected the reporting of occupational problems and will be avoided in further studies.

In spite of these limitations, our study adds significant information to the field of PCS treatment. We provide detailed data on relevant clinical variables that can be employed in larger studies in this field in the future. Our exploratory analysis of treatment effects supports the notion that online OT is feasible in PCS and that interactive online OT may be a promising treatment strategy for affected patients.

As such, we hope our data helps to pave the way for new, effective, and scalable treatment options to improve the situation of people with PCS.

## Supporting information

**S1 Table. Differences between study arms adjusted for baseline scores (cases with missing follow-up data excluded).**
(DOCX)

**S1 File. German version of study protocol and supplementary material and amendments.**
(PDF)

**S2 File. English translation of study protocol and supplementary material and amendments.**
(PDF)

**S3 File. CONSORT 2010 checklist.**
(PDF)

## Author contributions

**Conceptualization:** Georg M. N. Behrens, Alexandra Dopfer-Jablonka, Frank Müller, Christine Happle.

**Data curation:** Dominik Schröder, Andrea Stölting.

**Formal analysis:** Dominik Schröder, Andrea Stölting, Sandra Klawitter, Frank Klawonn, Frank Müller.

**Funding acquisition:** Alexandra Dopfer-Jablonka, Frank Müller.

**Investigation:** Andrea Stölting, Christina Müllenmeister, Aisha Cook, Nadja Wegner, Tim Schmachtenberg.

**Methodology:** Georg M. N. Behrens, Alexandra Dopfer-Jablonka.

**Project administration:** Andrea Stölting, Christine Happle.

**Supervision:** Christine Happle.

**Writing – original draft:** Dominik Schröder, Andrea Stölting, Christine Happle.

**Writing – review & editing:** Dominik Schröder, Andrea Stölting, Christina Müllenmeister, Georg M. N. Behrens, Martin Wetzke, Tim Schmachtenberg, Alexandra Dopfer-Jablonka, Frank Müller, Christine Happle.

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
