## [Decision Letter · Decision Letter 0]

11 Feb 2025

PONE-D-24-37522Improvement in quality of life and cognitive function in Post Covid Syndrome after online occupational therapy: results from a randomized controlled pilot studyPLOS ONE

Dear Dr. Stölting,

Thank you for submitting your manuscript to PLOS ONE. After careful consideration, we feel that it has merit but does not fully meet PLOS ONE’s publication criteria as it currently stands. Therefore, we invite you to submit a revised version of the manuscript that addresses the points raised during the review process.

We look forward to receiving your revised manuscript.

Kind regards,

Yalong Dang

Academic Editor

PLOS ONE

Journal Requirements:

The project received funding through the German Federal Ministry for Education and Research 

(Grant number 01EP2103C)

5. In the online submission form you indicate that your data is not available for proprietary reasons and have provided a contact point for accessing this data. Please note that your current contact point is a co-author on this manuscript. According to our Data Policy, the contact point must not be an author on the manuscript and must be an institutional contact, ideally not an individual. Please revise your data statement to a non-author institutional point of contact, such as a data access or ethics committee, and send this to us via return email. Please also include contact information for the third party organization, and please include the full citation of where the data can be found.

6. Please amend your list of authors on the manuscript to ensure that each author is linked to an affiliation. Authors’ affiliations should reflect the institution where the work was done (if authors moved subsequently, you can also list the new affiliation stating “current affiliation:….” as necessary).

7. Please include a new copy of Table 1, 2, 3, Supplementary Table 1 in your manuscript; the current table is difficult to read. Please follow the link for more information: https://blogs.plos.org/plos/2019/06/looking-good-tips-for-creating-your-plos-figures-graphics/

8. Please remove all personal information, ensure that the data shared are in accordance with participant consent, and re-upload a fully anonymized data set. 

Additional Editor Comments :

Reviewer 1: Line 70: to state ‘explore efficacy ‘ as 'efficacy' is generally not the primary focus of feasibility study.

Line 72: the word ‘underpowered’ is to be removed since no sample size was calculated.

Line 75: typo ‘The aim of the here’.

Line 102, 105-107: a separate list of exclusion criteria is to be provided. The participants were required to fill in the assessment tools/inventories/questionnaires, and some of the tools may require some level of understanding, either in English or German. Is the level of education one of the criteria?

This language of communication/used in this study is to be clearly stated, e.g., for emails, questionnaires, interviews, participant information sheets, consent forms, etc.

Line 129-137,146-147: the language and mode of administration of the tools/questionnaires are to be stated.

Line 139-141: the sentence requires revision.

Line 150: the reason for using the Wilcoxon signed-ranked test is to be stated.

Line 151-153: the exact name of the statistical test is to be stated.

Line 154-155: the information on the missing values/data is to be provided, e.g., percentage missing data, the pattern of missing data, etc. Describe the application of LOCF considering the characteristics of the dropout participants, e.g., any changes in the outcome, etc. If there is a significant number of dropout participants during follow-up, it is recommended to use the multiple imputation method.

Line 155-156: sentence ‘As sensitivity analysis adjusted between-group differences were reported with missing data excluded’ requires revision.

Line 157: the level of the confidence interval is to be stated e.g. 95%.

Line 158: one or two-tailed test is to be stated.

Ensure all the statistical tests mentioned in the results section are stated in the methodology statistical analysis section.

Ensure the decimal points for the p values are consistent throughout the manuscript.

Line 186: typo p 0.04 (e.g. p=0.04)

Line 188: typo p 0.003 (e.g. p=0.003)

Line 190: typo p 0.01 (e.g. p=0.01)

Table 1, Table 2, Supplementary Table 1 & Table 3: the title is to be placed on top of the table.

Line 192-193 Table 1: male gender is to be presented apart from female. The table looks confusing. On the top, it was stated as total n, but there were figures missing in the variables column (left). For reader convenience, it is best to include the sample size (n) for each variable.

Apart from intent to treat analysis, per protocol analysis is to be presented.

Line 224-233 Table 2 & Supplementary Table 1: were the data normally distributed? The distribution assumption of the data should be specified. If the data is skewed, it requires transformation or applying an alternative statistical test.

Line 250-251 Table 3: the table is blurry. Avoid using screenshot table(s). If possible, all tables are to be drawn directly within the manuscript.

Line 254: Wilcoxon signed-rank test

Table 2, Supplementary Table 1 and Table 3: decimal points are to be standardized where applicable.

Effect size could be presented.

Some references did not conform to the journal format.

Reviewer 2: The manuscript titled "Feasibility and Efficacy of Online Occupational Therapy in Post-COVID Syndrome: A Pilot Randomized Controlled Trial" presents novel exploratory results regarding digital interventions for Post-COVID Syndrome (PCS).

Limitations: 1. significant baseline variability (age, sex, memory function) was not fully controlled or adjusted. 2. High dropout rates (e.g., 53.8% in the video group) challenge the internal validity and raise concerns about participant engagement. 3. Sample Size Limitation: No a priori power calculation was carried out, although the authors explianed this concern in Discussion. 4. Statistical analysis: methods such as "last observation carried forward" (LOCF) need justification given group differences and missing data.

Please revise the MS on the following aspects:1. Address baseline imbalances (e.g., adjust for age and sex). 2. Clarify objectives and align outcomes with hypotheses. 3. Strengthen statistical rigor with detailed subgroup analyses and fine-tune methodological transparency.

Reviewers' comments:

Reviewer's Responses to Questions

**Comments to the Author**

1. Is the manuscript technically sound, and do the data support the conclusions?

Reviewer #1: Partly

Reviewer #2: Yes

2. Has the statistical analysis been performed appropriately and rigorously? 

Reviewer #1: No

Reviewer #2: Yes

3. Have the authors made all data underlying the findings in their manuscript fully available?

Reviewer #1: Yes

Reviewer #2: Yes

4. Is the manuscript presented in an intelligible fashion and written in standard English?

Reviewer #1: Yes

Reviewer #2: Yes

5. Review Comments to the Author

Reviewer #1: Line 70: to state ‘explore efficacy ‘ as 'efficacy' is generally not the primary focus of feasibility study.

Line 72: the word ‘underpowered’ is to be removed since no sample size was calculated.

Line 75: typo ‘The aim of the here’.

Line 102, 105-107: a separate list of exclusion criteria is to be provided. The participants were required to fill in the assessment tools/inventories/questionnaires, and some of the tools may require some level of understanding, either in English or German. Is the level of education one of the criteria?

This language of communication/used in this study is to be clearly stated, e.g., for emails, questionnaires, interviews, participant information sheets, consent forms, etc.

Line 129-137,146-147: the language and mode of administration of the tools/questionnaires are to be stated.

Line 139-141: the sentence requires revision.

Line 150: the reason for using the Wilcoxon signed-ranked test is to be stated.

Line 151-153: the exact name of the statistical test is to be stated.

Line 154-155: the information on the missing values/data is to be provided, e.g., percentage missing data, the pattern of missing data, etc. Describe the application of LOCF considering the characteristics of the dropout participants, e.g., any changes in the outcome, etc. If there is a significant number of dropout participants during follow-up, it is recommended to use the multiple imputation method.

Line 155-156: sentence ‘As sensitivity analysis adjusted between-group differences were reported with missing data excluded’ requires revision.

Line 157: the level of the confidence interval is to be stated e.g. 95%.

Line 158: one or two-tailed test is to be stated.

Ensure all the statistical tests mentioned in the results section are stated in the methodology statistical analysis section.

Ensure the decimal points for the p values are consistent throughout the manuscript.

Line 186: typo p 0.04 (e.g. p=0.04)

Line 188: typo p 0.003 (e.g. p=0.003)

Line 190: typo p 0.01 (e.g. p=0.01)

Table 1, Table 2, Supplementary Table 1 & Table 3: the title is to be placed on top of the table.

Line 192-193 Table 1: male gender is to be presented apart from female. The table looks confusing. On the top, it was stated as total n, but there were figures missing in the variables column (left). For reader convenience, it is best to include the sample size (n) for each variable.

Apart from intent to treat analysis, per protocol analysis is to be presented.

Line 224-233 Table 2 & Supplementary Table 1: were the data normally distributed? The distribution assumption of the data should be specified. If the data is skewed, it requires transformation or applying an alternative statistical test.

Line 250-251 Table 3: the table is blurry. Avoid using screenshot table(s). If possible, all tables are to be drawn directly within the manuscript.

Line 254: Wilcoxon signed-rank test

Table 2, Supplementary Table 1 and Table 3: decimal points are to be standardized where applicable.

Effect size could be presented.

Some references did not conform to the journal format.

Reviewer #2: The manuscript titled "Feasibility and Efficacy of Online Occupational Therapy in Post-COVID Syndrome: A Pilot Randomized Controlled Trial" presents novel exploratory results regarding digital interventions for Post-COVID Syndrome (PCS).

Limitations: 1. significant baseline variability (age, sex, memory function) was not fully controlled or adjusted. 2. High dropout rates (e.g., 53.8% in the video group) challenge the internal validity and raise concerns about participant engagement. 3. Sample Size Limitation: No a priori power calculation was carried out, although the authors explianed this concern in Discussion. 4. Statistical Rigor: methods such as "last observation carried forward" (LOCF) need justification given group differences and missing data.

Please revise the MS on the following aspects:1. Address baseline imbalances (e.g., adjust for age and sex). 2. Clarify objectives and align outcomes with hypotheses. 3. Strengthen statistical rigor with detailed subgroup analyses and fine-tune methodological transparency.

6. PLOS authors have the option to publish the peer review history of their article (what does this mean? ). If published, this will include your full peer review and any attached files.

**Do you want your identity to be public for this peer review?** For information about this choice, including consent withdrawal, please see our Privacy Policy .

Reviewer #1: No

Reviewer #2: No

---

## [Author Response · Author response to Decision Letter 1]

21 Mar 2025

Reviewer 1

Firstly, we would like to thank the reviewer for their time and effort. Their comments helped us to significantly improve our work!

Line 70: to state ‘explore efficacy ‘ as 'efficacy' is generally not the primary focus of feasibility study.

We agree to the reviewers notion and accordingly stated the exploratory nature of this part of our analysis in the revised version (line 70).

Line 72: the word ‘underpowered’ is to be removed since no sample size was calculated.

We removed this word from line 72.

Line 75: typo ‘The aim of the here’.

We rephrased this paragraph (lines 76 ff)

Line 102, 105-107: a separate list of exclusion criteria is to be provided. The participants were required to fill in the assessment tools/inventories/questionnaires, and some of the tools may require some level of understanding, either in English or German. Is the level of education one of the criteria?

This language of communication/used in this study is to be clearly stated, e.g., for emails, questionnaires, interviews, participant information sheets, consent forms, etc.

This is an important point, and we thank the reviewer for raising it. All communication was provided in German language only. We now added this information to the methodology of our paper (lines 104 ff).

Line 129-137,146-147: the language and mode of administration of the tools/questionnaires are to be stated.

Again, we would like to thank the reviewer for pointing out this lacking information, which was added to the methodology in the revised version (lines 104 ff).

Line 139-141: the sentence requires revision.

We revised the sentence (line 142-143).

Line 150: the reason for using the Wilcoxon signed-ranked test is to be stated.

Within-group outcomes did not meet normality assumptions, and we applied Wilcoxon signed-rank tests (Holm-adjusted) for changes over time (T1, T2 & T3) within each study arm. We now revised the methodology section (lines 152 ff).

Line 151-153: the exact name of the statistical test is to be stated.

We revised the method section and stated all exact names (lines 152 ff).

Line 154-155: the information on the missing values/data is to be provided, e.g., percentage missing data, the pattern of missing data, etc. Describe the application of LOCF considering the characteristics of the dropout participants, e.g., any changes in the outcome, etc. If there is a significant number of dropout participants during follow-up, it is recommended to use the multiple imputation method.

We would like to thank the reviewer for this comment. We followed these recommendations and changed the statistical analysis. Changes are as follows:

• we imputed missing values using multiple imputation,

• we used a linear mixed model

• we adjusted for a possible non-linear relationship between outcomes and baseline values of outcomes using B-splines (to capture potential nonlinearity; see e.g. 10.1186/s12874-016-0141-3)

We describe all changes and the methodological approach in detail in the revised methods section (lines 152 ff).

Line 155-156: sentence ‘As sensitivity analysis adjusted between-group differences were reported with missing data excluded’ requires revision.

We rephrased the method section accordingly (lines 152 ff).

Line 157: the level of the confidence interval is to be stated e.g. 95%.

Done!

This is a good suggestion, we report 95% CI for inter-group analyses. Please refer to the lines 152 ff for the revised methodology section.

Line 158: one or two-tailed test is to be stated.

We added this important information to the text (All statistical tests were two-sided). Please refer to the lines 152 ff for the revised methodology section.

Ensure all the statistical tests mentioned in the results section are stated in the methodology statistical analysis section.

We agree that the statistical methodology could be described in more detail and added the statistical tests we used to compare patient characteristics and baseline values between groups: Sample characteristics were characterized using descriptive and bivariate statistics (Kruskal-Wallis test for continuous variables and Fisher-Freeman-Halton test for categorial variables). Please refer to the lines 152 ff for the revised methodology section.

Ensure the decimal points for the p values are consistent throughout the manuscript.

Line 186: typo p 0.04 (e.g. p=0.04)

Line 188: typo p 0.003 (e.g. p=0.003)

Line 190: typo p 0.01 (e.g. p=0.01)

We harmonized the decimal points and reporting of p values throughout the manuscript.

Table 1, Table 2, Supplementary Table 1 & Table 3: the title is to be placed on top of the table.

We placed all titles on top of the tables.

Line 192-193 Table 1: male gender is to be presented apart from female. The table looks confusing. On the top, it was stated as total n, but there were figures missing in the variables column (left). For reader convenience, it is best to include the sample size (n) for each variable.

We thank the reviewer for this point and now reformatted the table so that all items are listed in logical order. This will help readers to better understand the characteristics of our cohort (line 198-199).

Apart from intent to treat analysis, per protocol analysis is to be presented.

We thank the reviewer for this suggestion. In response, we now added multiple imputation to the approach, used a linear-mixed-effects model and added per protocol analysis and rephrased the method section accordingly (lines 152 ff).

Line 224-233 Table 2 & Supplementary Table 1: were the data normally distributed? The distribution assumption of the data should be specified. If the data is skewed, it requires transformation or applying an alternative statistical test.

The differences between the time points followed a normal distribution. Please refer to the lines 152 ff for the revised methodology section.

Line 250-251 Table 3: the table is blurry. Avoid using screenshot table(s). If possible, all tables are to be drawn directly within the manuscript.

We apologize for the blurriness. We now provide all tables within the text as original text/number-format.

Line 254: Wilcoxon signed-rank test

We thank the reviewer for pointing this out and changed the wording accordingly (lines 152 ff, revised statistical methodology).

Table 2, Supplementary Table 1 and Table 3: decimal points are to be standardized where applicable.

We harmonized the reporting in tables and text.

Effect size could be presented.

We appreciate the suggestion to present standardized effect sizes (e.g., Cohen’s d). In this pilot study, we chose to focus on clinically meaningful changes, particularly in relation to minimal clinically important differences, rather than relying on purely statistical standardizations of the outcome data. While effect sizes like Cohen’s d may help to compare results across different studies and scales, this does not necessarily translate directly into clinically relevant benefit for affected patients. We chose to report detailed group means and 95% confidence intervals, which allow readers to derive standardized effect sizes if needed. We believe this approach underscores the real-world impact of our findings and aligns with our aim of assessing the clinical significance of online OT in Post-COVID Syndrome.

Some references did not conform to the journal format.

We apologize for this and reformatted the references, hoping to have met all formal criteria required PLOS ONE.

Again, we would like to thank the reviewer for their time and work which helped us to improve our paper!

Reviewer 2: The manuscript titled "Feasibility and Efficacy of Online Occupational Therapy in Post-COVID Syndrome: A Pilot Randomized Controlled Trial" presents novel exploratory results regarding digital interventions for Post-COVID Syndrome (PCS).

Limitations: 1. significant baseline variability (age, sex, memory function) was not fully controlled or adjusted. 2. High dropout rates (e.g., 53.8% in the video group) challenge the internal validity and raise concerns about participant engagement. 3. Sample Size Limitation: No a priori power calculation was carried out, although the authors explained this concern in Discussion. 4. Statistical analysis: methods such as "last observation carried forward" (LOCF) need justification given group differences and missing data.

We thank the reviewer for their time and critical reading of our manuscript. The points raised are important, and they gave us a chance to further refine our statistical approach and to significantly improve our manuscript.

Please revise the MS on the following aspects:1. Address baseline imbalances (e.g., adjust for age and sex).

We thank the reviewer for this comment. However, our study was not designed for such post hoc analyses. Moreover, given the low variation in age and the overall high proportion of female participants in our trials, adjusting for these factors could introduce statistical issues such as overfitting, reduced statistical power, and potential collider bias. Overfitting may occur due to the inclusion of variables with minimal variance, while reduced power can result from unnecessary model complexity. We agree that this limitation needs further consideration in future studies and are in fact planning a follow up study in a larger patient group. We rephrased the respective part of our limitation paragraph in the discussion section to emphasize this part (lines 317-320).

2. Clarify objectives and align outcomes with hypotheses.

We thank the reviewer for pointing this out and now state the objective and hypothesis of our analysis in the last paragraph of the introduction (lines 76 ff). We also begin each section of the results section with a brief referral to the respective aim or objective.

3. Strengthen statistical rigor with detailed subgroup analyses and fine-tune methodological transparency.

Thank you for this comment which is in line with some suggestions of our first reviewer and gave us the chance to refine our statistical analyses and thoroughly revise the respective methodology section. We made the following key changes:

• We imputed missing values using multiple imputation

• Used a linear mixed model

• Adjusted for a possible non-linear relationship between outcomes and baseline values of outcomes using a by B-splines (to capture potential nonlinearity; 10.1186/s12874-016-0141-3)

Please refer to the lines 152 ff for the revised methodology section.

Again, we would like to thank the reviewers and the editorial team for their review and thoughtful suggestions that helped us to improve our manuscript!

---

## [Decision Letter · Decision Letter 1]

6 Apr 2025

PONE-D-24-37522R1Improvement in quality of life and cognitive function in Post Covid Syndrome after online occupational therapy: results from a randomized controlled pilot studyPLOS ONE

Dear Dr. Stölting,

Thank you for submitting your manuscript to PLOS ONE. After careful consideration, we feel that it has merit but does not fully meet PLOS ONE’s publication criteria as it currently stands. Therefore, we invite you to submit a revised version of the manuscript that addresses the points raised during the review process.

We look forward to receiving your revised manuscript.

Kind regards,

Yalong Dang

Academic Editor

PLOS ONE

Journal Requirements:

Reviewers' comments:

Reviewer's Responses to Questions

**Comments to the Author**

1. If the authors have adequately addressed your comments raised in a previous round of review and you feel that this manuscript is now acceptable for publication, you may indicate that here to bypass the “Comments to the Author” section, enter your conflict of interest statement in the “Confidential to Editor” section, and submit your "Accept" recommendation.

Reviewer #1: (No Response)

Reviewer #2: All comments have been addressed

2. Is the manuscript technically sound, and do the data support the conclusions?

Reviewer #1: Partly

Reviewer #2: Yes

3. Has the statistical analysis been performed appropriately and rigorously? 

Reviewer #1: No

Reviewer #2: Yes

4. Have the authors made all data underlying the findings in their manuscript fully available?

Reviewer #1: Yes

Reviewer #2: Yes

5. Is the manuscript presented in an intelligible fashion and written in standard English?

Reviewer #1: Yes

Reviewer #2: Yes

6. Review Comments to the Author

Reviewer #1: Minor comments:

Table 2 title: The abbreviation 'LOCF' is to be replaced.

Line 165: To state the assumptions for repeated measure ANOVA were fulfilled prior to use.

Reviewer #2: A nice job ready for publication after address the two minor points raised by Reviewer 1, thank you all authors.

7. PLOS authors have the option to publish the peer review history of their article (what does this mean? ). If published, this will include your full peer review and any attached files.

**Do you want your identity to be public for this peer review?** For information about this choice, including consent withdrawal, please see our Privacy Policy .

Reviewer #1: No

Reviewer #2: No

---

## [Author Response · Author response to Decision Letter 2]

9 Apr 2025

Reviewer #1:

We would like to thank reviewer #1 for their work in assessing our manuscript. We have addressed the remaining concerns as stated below:

Table 2 title: The abbreviation 'LOCF' is to be replaced.

We have replaced the abbreviation ‘LOCF’ with ‘multiple imputations’ to remove the need for an abbreviation in the title.

Line 165: To state the assumptions for repeated measure ANOVA were fulfilled prior to use.

We now clarify that we tested our data for conforming to the underlying assumptions on lines 165 and 166.

Reviewer #2: A nice job ready for publication after address the two minor points raised by Reviewer 1, thank you all authors.

We thank reviewer #2 for their help in assessing and improving our manuscript!

---

## [Decision Letter · Decision Letter 2]

15 Apr 2025

Improvement in quality of life and cognitive function in Post Covid Syndrome after online occupational therapy: results from a randomized controlled pilot study

PONE-D-24-37522R2

Dear Dr. Stölting,

We’re pleased to inform you that your manuscript has been judged scientifically suitable for publication and will be formally accepted for publication once it meets all outstanding technical requirements.

Kind regards,

Yalong Dang

Academic Editor

PLOS ONE

Additional Editor Comments (optional):

Reviewers' comments:

Reviewer's Responses to Questions

**Comments to the Author**

1. If the authors have adequately addressed your comments raised in a previous round of review and you feel that this manuscript is now acceptable for publication, you may indicate that here to bypass the “Comments to the Author” section, enter your conflict of interest statement in the “Confidential to Editor” section, and submit your "Accept" recommendation.

Reviewer #1: All comments have been addressed

2. Is the manuscript technically sound, and do the data support the conclusions?

Reviewer #1: (No Response)

3. Has the statistical analysis been performed appropriately and rigorously? 

Reviewer #1: (No Response)

4. Have the authors made all data underlying the findings in their manuscript fully available?

Reviewer #1: (No Response)

5. Is the manuscript presented in an intelligible fashion and written in standard English?

Reviewer #1: (No Response)

6. Review Comments to the Author

Reviewer #1: (No Response)

7. PLOS authors have the option to publish the peer review history of their article (what does this mean? ). If published, this will include your full peer review and any attached files.

**Do you want your identity to be public for this peer review?** For information about this choice, including consent withdrawal, please see our Privacy Policy .

Reviewer #1: No

---

## [Editor Report · Acceptance letter]

PONE-D-24-37522R2

PLOS ONE

Dear Dr. Stölting,

I'm pleased to inform you that your manuscript has been deemed suitable for publication in PLOS ONE. Congratulations! Your manuscript is now being handed over to our production team.

Kind regards,

on behalf of

Dr Yalong Dang

Academic Editor

PLOS ONE